# The Anticancer Activity Conferred by the Mud Crab Antimicrobial Peptide Scyreprocin through Apoptosis and Membrane Disruption

**DOI:** 10.3390/ijms23105500

**Published:** 2022-05-14

**Authors:** Ying Yang, Hui-Yun Chen, Hua Hao, Ke-Jian Wang

**Affiliations:** 1State Key Laboratory of Marine Environmental Science, College of Ocean and Earth Sciences, Xiamen University, Xiamen 361102, China; yangying0803@gmail.com (Y.Y.); hychen@xmu.edu.cn (H.-Y.C.); hhao@xmu.edu.cn (H.H.); 2State-Province Joint Engineering Laboratory of Marine Bioproducts and Technology, College of Ocean and Earth Sciences, Xiamen University, Xiamen 361102, China; 3Fujian Innovation Research Institute for Marine Biological Antimicrobial Peptide Industrial Technology, College of Ocean & Earth Sciences, Xiamen University, Xiamen 361102, China

**Keywords:** scyreprocin, antimicrobial peptide, apoptosis, anti-tumor, non-small-cell lung cancer

## Abstract

Scyreprocin is an antimicrobial peptide first identified in the mud crab *Scylla paramamosain*. Herein, we showed that its recombinant product (rScyreprocin) could significantly inhibit the growth of human lung cancer NCI-H460 cells (H460), but showed no cytotoxicity to human lung fibroblasts (HFL1). rScyreprocin was a membrane-active peptide that firstly induced the generation of reactive oxygen species (ROS) in H460, and led to endoplasmic reticulum stress and Ca^2+^ release, which resulted in mitochondrial dysfunction and subsequently activation of caspase-3 cascades, and ultimately led to apoptosis. The comprehensive results indicated that rScyreprocin exerted anticancer activity by disrupting cell membrane and inducing apoptosis. The in vivo efficacy test demonstrated that intratumoral injection of rScyreprocin significantly inhibited the growth of H460 xenografts, which was close to that of the cisplatin (inhibition rate: 69.94% vs. 80.76%). Therefore, rScyreprocin is expected to become a promising candidate for the treatment of lung cancer.

## 1. Introduction

Cancer is a major public health problem and the leading cause of death worldwide. With ~2.1 million cases in 2018, lung cancer is the most commonly diagnosed and fatal cancer for men and women worldwide [1]. Although the death rate of lung cancer has declined in some countries, the deaths caused by lung cancer have surpassed that of other major cancers combined in 2017 [2]. There are two main subtypes of lung cancer: small-cell lung carcinoma and non-small-cell lung carcinoma (NSCLC). NSCLC accounts for about 85% of total lung cancer cases [3]. For the patients with stage I, II, or IIIA NSCLC, surgery can be performed [4]. About 40% of newly diagnosed NSCLC patients are in stage IV. Due to the metastasis of NSCLC, the surgery is less effective; thus, chemotherapy and radiotherapy are usually recommended as the first-line treatment [5].

Although new treatments (e.g., immunotherapy) have been rapidly developed in recent years, platinum-based chemotherapy is still the preferred clinical treatment for NSCLC. Cisplatin (CDDP), one of the most commonly used platinum drugs, inhibits cancer cell division by covalently binding to the purine DNA bases, leading to cell apoptosis [6]. However, CDDP can also cause severe side effects [7]. Some patients are intrinsically resistant to CDDP-based therapies, while a number of patients acquire chemoresistance during therapy [8]. Research has proved that the use of doublets including a platinum and a third-generation agent are equally effective and widely adopted, which could improve the life quality of patients [9]. Therefore, exploration of bioactive molecules with potent anticancer activity and/or less cytotoxicity is the current trend in anticancer drug development. 

Antimicrobial peptides (AMPs) are important components of the innate immunity, and they exert activity against pathogens [10]. AMPs have become one of the most promising antibacterial agents against various antimicrobial-resistant bacteria. Intriguingly, the anticancer activities of some AMPs have been described, which are termed anticancer peptides (ACPs) [11]. ACPs, such as pleuricidins (from Atlantic flatfishes), Aurein 1.2 (from the frog *Litoria aurea*), and human neutrophil peptide-1, have been shown to have high efficacy in killing cancer cells [12,13,14]. However, the anticancer mechanisms of AMPs have not yet been fully illustrated. They appear to include plasma membrane disruption [15], activation of lysosomal death pathway [16], and apoptosis [17]. 

Scyreprocin is a cationic AMP that was first identified in the mud crab *Scylla paramamosain* [18,19]. The recombinant product of scyreprocin (rScyreprocin) exhibits potent inhibitory activity against various strains of bacteria, fungi, fungal biofilms, and spore germination and can improve fish survival rate under bacterial infection [18]. rScyreprocin exerts high germicidal activity by destroying microbial membranes. In addition, it can induce apoptosis of fungal cells [18]. Therefore, it was speculated that scyreprocin might have anticancer activity. In this study, we investigated the in vitro anticancer activity and the action mechanism of rScyreprocin. Moreover, a xenograft model was used to evaluate the in vivo anticancer activity of rScyreprocin to explore its future therapeutic application prospects.

## 2. Results

### 2.1. Recombinant scyreprocin (rScyreprocin) Inhibited Proliferation, Migration, and Colony Formation Ability of Cancer Cells but Not Non-Cancer Cells

rScyreprocin was expressed, purified, and stored at −80 °C before use (Appendix A). The results showed that rScyreprocin inhibited the growth of H460 (NSCLC) and HepG2 (liver cancer), but was non-toxic to cell lines of normal origin (lung fibroblasts HFL1 and liver cells L02) (Figure 1A). rScyreprocin also inhibited the growth of several tested cancer cell lines such as T24 (bladder cancer) and Du145 (prostate cancer) cells. rScyreprocin had an inhibitory effect on HeLa (cervical cancer) cells in the concentration range tested in this study (up to 20 μM), but this was not statistically significant. rScyreprocin treatment (5 μM) inhibited the colony formation ability of H460, HepG2, T24, HeLa, and Du145 cells, but had no effect on HFL1 and L02 cells (Figure 1B). These results suggested that rScyreprocin had specific inhibitory activity against cancer cells. 

As shown in Figure 2A, rScyreprocin exerted an inhibitory effect on the H460 cell viability, but had no effect on HFL1 cells. The IC50 values of rScyreprocin at 24, 48, and 72 h (18.91 μM, 11.70 μM, and 6.27 μM, respectively) were lower than that of CDDP (31.91 μM, 22.71 μM, and 16.20 μM, respectively) (Figure 2A). Therefore, rScyreprocin had no cytotoxicity towards HFL1 cells and had potent anticancer activity against H460 cells with low effective concentrations.

As shown in Figure 2B, rScyreprocin treatment resulted in fewer H460 colonies while showing no inhibitory effect against HFL1 cells (Figure 2B). Moreover, rScyreprocin inhibited the migration ability of H460 cells (Figure 2C). Cells with abnormal morphologies, such as formation of apoptotic corpuscle and cell necrosis, were observed at the edge of the scratch in rScyreprocin-treated samples (Appendix A). This observation suggested that high concentrations of rScyreprocin induced cell death, which further led to larger scratch widths.

### 2.2. rScyreprocin Induced Apoptosis and Membrane Disruption of Cancer Cells

In the CDDP treatment group, H460 and HFL1 cells showed classic apoptosis features (Figure 3A,B). H460 cells treated with rScyreprocin not only showed classic apoptosis characteristics similar to those treated with CDDP, but also showed a certain concentration-dependent cell structural destruction and leakage of cell contents (Figure 3A). HFL1 cells treated with rScyreprocin showed no morphological changes (Figure 3B). rScyreprocin induced apoptosis in H460 cells (Figure 3C), but not in HFL1 cells (Figure 3C). These results suggested that rScyreprocin exerted its anticancer activity by both inducing cell apoptosis and disrupting cell membrane integrity (possibly cell necrosis). 

### 2.3. rScyreprocin Entered into Cells and Distributed in Organelles

rScyreprocin could enter both H460 and HFL1 cells (Figure 4A), and was detected in all cell fractions (Figure 4B and Appendix A). rScyreprocin was co-located with ER and lysosomes of H460 cells (Figure 4C). TEM observation showed the location of rScyreprocin in rough endoplasmic reticulum (RER), mitochondria, cytoplasm, and nucleolus (Figure 4D). In rScyreprocin-treated HFL1 cells, rScyreprocin signals were only seen in the cytoplasm, and not localized to specific organelles (Figure 4F). 

In rScyreprocin-treated H460 cells, significant morphological changes in organelles (e.g., mitochondria, Golgi apparatus, and RER) were observed (Figure 4E). Meanwhile, rScyreprocin was detected in the cytoplasm of HFL1 (Figure 4F), and the organelles in rScyreprocin-treated HFL1 cells showed no morphological changes (Figure 4G).

### 2.4. rScyreprocin Induced Endoplasmic Reticulum (ER) Stress and Apoptosis in H460 Cells

As shown in Figure 5, in rScyreprocin-treated H460 cells, the expression level of pro-apoptotic proteins increased, and the expression level of anti-apoptotic proteins decreased. The expression levels of endoplasmic reticulum stress (ER-stress) marker proteins, eg, such as C/EBP homologous protein (CHOP) and activating transcription factor 4 (ATF-4), were significantly increased. Meanwhile, rScyreprocin treatment had no significant effect on the expression level of the related proteins in HFL1.

BAPTA (1,2-bis (2-aminophenoxy) ethane-N,N,N′,N′-tetraacetic acid tetrakis) is a calcium-selective chelator, and N-acetylcysteine (NAC) is an inhibitor of reactive oxygen species (ROS). Pretreatment of BAPTA and NAC could inhibit the increase of intracellular Ca^2+^ and ROS concentration, respectively. 

When treated with rScyreprocin for 8 and 10 h, the intracellular ROS concentration of H460 cells increased, while a remarkable intracellular Ca^2+^ concentration elevation between 8 and 12 h after rScyreprocin treatment was observed (Figure 6A). In the presence of NAC, the viability of rScyreprocin-treated H460 cells increased (Figure 6B), and no apoptotic effect was detected (Figure 6C). Pretreatment of NAC but not BAPTA/AM prevented rScyreprocin-induced ROS production (Figure 6D). In addition, the increase in the intracellular Ca^2+^ concentration ([Ca^2+^]_i_) was blocked in NAC-pretreated cells (Figure 6E). Therefore, rScyreprocin first induced ROS production and accumulation, which further led to the [Ca^2+^]_i_ increase in H460 cells.

After rScyreprocin treatment, the proportion of H460 cells with decreased mitochondrial membrane potential (MMP) (36.61 ± 0.42%) was higher compared to the control group (9.41 ± 0.33%). NAC pretreatment partially prevented rScyreprocin-induced MMP reduction (23.77 ± 1.74%) (Figure 6F). These results indicated that MMP loss occurred downstream of ROS production. 

The rScyreprocin-treated H460 cells contained a higher level of Bcl-2 associated X protein (Bax) in mitochondrial factions, and a parallel decrease in cytosolic fractions. Meanwhile, rScyreprocin also induced an increase in cytosolic cytochrome c. NAC pretreatment in turn prevented the translocation of Bax, Bcl-2, and cytochrome c induced by rScyreprocin (Figure 6G, Appendix A). These data indicated that the rScyreprocin-induced intracellular ROS accumulation triggered the redistribution of components in the caspase-3-dependent apoptotic pathway, leading to mitochondrial dysfunction and cell apoptosis. However, rScyreprocin showed no similar effects on HFL1 cells (Appendix A).

### 2.5. rScyreprocin Exerted Anti-Tumor Effect in Nude Mice Model

The in vivo anticancer activity of rScyreprocin was evaluated. From the 15th day after administration, the difference of relative tumor volume (RTV) between the rScyreprocin-treatment group and the phosphate buffer saline (PBS) control became obvious (Figure 7A). The body weight of the mice showed no significant changes during the treatment (Figure 7B). The mice were sacrificed, and the tumor tissues were dissected (Figure 7C). The tumor growth inhibition rates of 3 mg kg^−1^ CDDP and 1.8 mg kg^−1^ rScyreprocin were 81.92% and 71.11%, respectively (Figure 7D). In the PBS-treated group, tumor tissues showed almost no necrosis and apoptosis (Figure 7E). In the CDDP- and rScyreprocin-treated groups, significant necrosis and apoptosis were observed in the tumor sections (Figure 7E). Compared with the PBS-treated group, significant reductions in Ki-67^+^ area and CD31^+^ area were observed in rScyreprocin-treated tumors (Figure 7F).

## 3. Discussion

Surgery and chemotherapy are common treatments for cancer. Postoperative infections caused by drug-resistant microorganisms and chemotherapy resistance of the cancer cells are the two main problems for treatment failure. If left unresolved, the rapid emergence of antimicrobial resistance and the rising cancer incidence may eventually lead to an estimated ~11.2 million deaths by 2050 [20,21]. Peptides with dual antimicrobial and anticancer activities are ideal candidates for the treatments of both cancer and drug-resistant microorganisms. In our previous study, rScyreprocin showed potent antimicrobial activity [18]. rScyreprocin can interact with and destroy the negatively charged microbial membranes, leading to cell lysis (in bacteria and fungi) or apoptosis (in fungi) [18]. Whether rScyreprocin might also possess anticancer activity attracts us toward further investigation. In this study, we first demonstrated that rScyreprocin exerted potent anticancer activity in vitro and in vivo, and further elucidated that rScyreprocin inhibited the growth of H460 cells through disrupting membranes and inducing apoptosis. 

The most common feature of AMPs with anticancer properties are α-helical, β-sheet, and extended AMPs [22,23,24]. These AMPs have been reported to first interact with the cancer cell membrane through electrostatic attraction, and then kill the cancer cell through membrane disruption [11]. Some could induce apoptosis [25], activate complements [26], and affect intracellular targets [27]. Scyreprocin is a cationic, cell-penetrating peptide that forms α-helical structures. Certain peptides are reported to enter the cell membrane through endocytosis or by the formation of transient pores in the membrane with the help of hydrophobic α-helical structures [28]. rScyreprocin induced significant membrane damage in H460 cells in a dose-dependent manner, but not in HFL1 cells (Figure 3). This selective anticancer activity of rScyreprocin may be related to the fact that the high asymmetry of the charge in the membrane of cancer cells would lead to higher rigidity and instability than that of non-cancer cells [29]. However, the structure of rScyreprocin and its dynamic interaction pattern with phospholipid bilayer deserve further elucidation. 

Some cancer chemotherapeutic drugs induce apoptosis in part through ROS generation [30]. ROS was an early signal that mediated the rScyreprocin-induced apoptosis, and the elevation of [Ca^2+^]_i_ occurred shortly after ROS generation. Intracellular Ca^2+^ plays a central role in regulating and sensing key cellular processes, including apoptosis, autophagy, and unfolded protein response (UPR) [31]. How did rScyreprocin-induced ROS generation lead to an increase in [Ca^2+^]_i_? Cells maintain Ca^2+^ homeostasis under normal conditions, and intracellular Ca^2+^ is mainly stored in organelles, such as ER and mitochondria [32]. The destruction of organelle membrane structure by rScyreprocin may be responsible for the [Ca^2+^]_i_ elevation. ROS and intracellular Ca^2+^ are two cross-talking messengers in various cellular process; disorder of either could lead to ER stress and apoptosis [33]. ER-stress events may trigger the accumulation of misfolded or unfolded proteins in ER lumen, resulting in increased cytoplasmic Ca^2+^ and disrupting cell homeostasis, leading to UPR activation to restore equilibrium of the ER [34]. In this study, rScyreprocin activated ER-stress markers (ATF-4 and CHOP), indicating that rScyreprocin induced ER stress in H460 cells (Figure 5). The exact role of Ca^2+^ signaling in rScyreprocin-induced ER stress remains to be studied. 

The significance of the ER-mitochondria interactions in controlling cellular functions is an emerging research topic, and Ca^2+^ is one of the main mediators of this inter-organelle communication [35]. Under ER stress, Ca^2+^ released from ER is promptly taken up by mitochondria. Low cytosolic Ca^2+^ increases can generate much higher mitochondrial Ca^2+^ peaks, which could lead to Ca^2+^ overload in mitochondria, the release of pro-apoptotic factors, and the activation of the apoptotic cascade [36]. Mitochondrial membrane depolarization has been detected as an early event of apoptosis [37]. After rScyreprocin treatment, the MMP decreased and [Ca^2+^]_i_ increased, implying mitochondrial dysfunction (Figure 6). Depolarization of MMP induced by rScyreprocin was related to the redistribution of Bax and Bcl-2, the release of cytochrome c from the mitochondria to the cytosol, and the cleavage of caspase-3 and poly (ADP-ribose) polymerases (PARP-1), and ultimately led to apoptosis (Figure 6). It should be noted that in our study, even though the above-mentioned protein translocation and activation were inhibited when H460 cells were pretreated with NAC or BAPTA in our study, rScyreprocin still caused apoptosis in BAPTA-pretreated H460 cells. These facts suggest that both ROS generation and Ca^2+^ overload contribute to the loss of MMP, and the apoptosis induced by the accumulated ROS may involve Ca^2+^ overload-dependent and -independent mechanisms.

AMPs with anticancer activity provide new strategies for cancer therapy; moreover, they might be the only class of compounds effective against multi-drug resistance infections as well as cancers. The anti-tumor efficacy of several AMPs has been evaluated in xenograft models and showed promising results. For instance, cecropin B1 showed a higher tumor growth inhibition effect than docetaxel in H460 xenografted mice [38], whilst the R-Tf-D-LP4 peptide induced apoptosis in subcutaneous HepG2 cell xenograft models [39]. The most representative AMP under clinical trial is LL-37, which has been under testing in a phase I/II trial to evaluate its efficacy against melanoma (NCT02225366). The in vivo results in this study showed that rScyreprocin exerted a promising inhibitory effect on the growth of H460 xenografts by inducing apoptosis and suppressing the proliferation of tumor cells. The consistency of the results obtained from in vivo and in vitro experiments suggested that rScyreprocin, as a novel anticancer peptide, would be valuable for further research and potential therapeutic application. 

Taken together, rScyreprocin was observed to be cell penetrating and have potent anticancer activity both in vitro and in vivo. Direct disruption of cell membranes and induction of apoptosis were the two main anticancer mechanisms of rScyreprocin. rScyreprocin-induced ROS generation led to ER stress and Ca^2+^ release, which further caused mitochondria dysfunction. The translocation of mitochondrial membrane proteins resulted in the loss of MMP, activation of caspase-3 cascades, and eventually led to cell apoptosis. rScyreprocin might be a promising anticancer agent for future application.

## 4. Materials and Methods

### 4.1. Reagents and Antibodies

CellTiter 96™ AQ_ueous_ One Solution Cell Proliferation Assay Kit was obtained from Promega (Madison, WI, USA). Qproteome™ Mitochondria isolation Kit was obtained from Qiagen (Valencia, CA, USA). ProteoExtract^®^ Subcellular Proteome Extraction Kit was purchased from Merck (Darmstadt, Germany). In situ Cell Death Detection Kit was purchased from Roche (Mannheim, Germany). Goat anti-Mouse IgG (H+L) secondary antibody (Dylight488), Goat anti-Rabbit IgG (H+L) secondary antibody (Dylight 650), ER-Tracker™ Green, LysoTracker™ Red DND-99, MitoTracker™ Orange CMTMRos, glutaraldehyde, Fluo-4/AM, 1,2-bis (2-aminophenoxy) ethane-N,N,N′,N′-tetraacetic acid tetrakis (acetoxymethyl ester; BAPTA/AM), N-acetylcysteine (NAC), 4′,6-diamidino-2-phenylindole dihydrochloride (DAPI), JC-1 Mitochondrial Potential Sensor, ATP determination Kit, cisplatin (CDDP), and 2′-7′-dichlorodihydropluorescein diacetate (DCFH-DA) were purchased from Thermo Fisher Scientific (Waltham, MA, USA). Golgi-Tracker Green (BODIPY^®^ FL C5-Ceramide) and iFluor™ 555 phalloidin were obtained from Yeasen Biotechnology (Shanghai, China). Goat-anti-Rabbit IgG (H+L) EM Grade 15 nm was obtained from Electron Microscopy Sciences (Fort Washington, PA, USA). The ECL Western Kit was obtained from Millipore Corporation (Billerica, MA, USA). Antibodies for PARP-1 (Cat# 9532), cleaved PARP-1 (Cat# 9541), Bax (Cat# 5023), Bcl-2 (Cat# 4223), pro-Caspase 3 (Cat# 9662), active-Caspase 3 (Cat# 9664), cytochrome c (Cat# 4280), β-actin (Cat# 3700), CD31 (Cat# 77699), and Ki-67 (Cat# 9449) were purchased from Cell Signaling Technology (Beverly, MA, USA).

### 4.2. Recombinant Protein and Antibody Preparation

The recombinant scyreprocin (rScyreprocin) and the scyreprocin antibody were prepared as previously described in [18]. 

### 4.3. Cell Lines and Cell Culture

Human non-small cell lung cancer (NSCLC) NCI-H460 cells (H460, Cat# SCSP-584, identifier CSTR:19375.09.3101HUMSCSP584) and human embryonic lung fibroblasts (HFL1, Cat# SCSP-5049, identifier CSTR:19375.09.3101HUMSCSP5049) were obtained from National Collection of Authenticated Cell Cultures, Chinese Academy of Sciences (Shanghai, China). Cells were cultured in a humidified atmosphere at 37 °C in 5% CO_2_.

### 4.4. Colony Formation

H460 (~300 cells well^−1^) and HFL1 (~100 cell well^−1^) cells were seeded in 6-well plates. After 10 h of incubation, rScyreprocin was added (0, 1, and 4 μM). After 48 h, the medium containing rScyreprocin was replaced with fresh medium. After an additional 12 days of incubation, the colonies were fixed with 4% (*w*/*v*) phosphate-buffered paraformaldehyde and stained with crystal violet. Colonies that contained > 50 cells were counted. The experiment was carried out in triplicate. 

### 4.5. Cell Viability Assay

H460 cells and HFL1 cells were seeded in 96-well plates at a density of 2 × 10^4^ cells well^−1^ and grown to ~70% confluence before being treated with rScyreprocin (0, 1, 2, 4, 8, or 16 μM) and CDDP (0, 8, 16, 32, 64, 128, and 256 μM), respectively. Bovine serum albumin (BSA, 16 μM) was used as a negative protein control. At different time points after incubation, cell viability was assessed by CellTiter 96™ AQ_ueous_ One Solution Cell Proliferation Assay Kit, and the results were obtained by a microplate reader (TECAN GENios; Tecan Group Ltd., Männedorf, Switzerland). The experiments were performed in triplicate and carried out under three different occasions (*n* = 3) with rScyreprocin expressed in different batches. The curve fit plots were generated using GraphPad Prism Software (version 5.01; GraphPad Software Inc., San Diego, CA, USA). Half maximal inhibitory concentration (IC_50_) was calculated by IBM SPSS statistics (version 22; IBM Corp., Armonk, NY, USA).

### 4.6. Electron Microscopy Observation

H460 and HFL1 cells were seeded in 48-well plates at a density of 2 × 10^5^ cells well^−1^. Cells were immersed in culture medium supplemented with 0, 1, 4, or 8 μM rScyreprocin; 8 μM BSA; or 16 μM CDDP for 24 h, respectively. For scanning electron microscope (SEM) observation, cells were rinsed with Hank’s balanced salt solution (HBSS; HyClone, Logan, UT, USA), fixed in pre-cooled 2.5% (*v*/*v*) phosphate-buffered glutaraldehyde at 4 °C for 2 h, dehydrated and gold-coated as per previous descriptions before being observed by a Zeiss Spura™ 55 Scanning Electron Microscope (Carl Zeiss Microscopy GmbH, Oberkochen, Germany) [40]. For transmission electron microscopy (TEM) observation, cells were digested with 0.25% (*w*/*v*) trypsin and collected by centrifugation. Cells were fixed in 2.5% (*v*/*v*) glutaraldehyde (for cell ultrastructural observation) or 4% (*w*/*v*) phosphate-buffered paraformaldehyde (for immune-colloidal gold labeling) at 4 °C for 5 h, washed with ice-cold PBS (pH 7.4) and pre-embedded in agarose. For ultrastructural observation, ultrathin sectioning and negative staining were performed following standard protocols [41]. For immuno-colloidal gold labeling, samples were blocked with 2% (*v*/*v*) normal goat serum (NGS, prepared in PBS) for 30 min, and incubated overnight with scyreprocin antibody (1:100, prepared in 0.5% NGS) at 4 °C. The scyreprocin antibody was recognized by Goat-anti-Rabbit IgG (H+L) EM Grade 15 nm. The sections were post-fixed with 4% (*w*/*v*) phosphate-buffered paraformaldehyde before being subjected to negative staining [41]. The samples were observed by a TEM (FEI Tecnai G2 F20; Eindhoven, The Netherlands).

### 4.7. Scratch-Wound Assay

H460 cells were seeded in 6-well plates and grown to a confluence monolayer. A scratch wound was inflicted with a p20 pipette tip. Plates were rinsed with HBSS to remove cell debris, and incubated with medium supplemented with rScyreprocin (0, 1, 2, 4, 8, and 16 μM). The wound closure was observed and imaged with an optical microscope at different time points (0, 3, 6, 9, 24, and 48 h). The experiment was carried out on three different occasions (*n* = 3).

### 4.8. Cell Apoptosis Detection

Cell apoptosis was assayed by the terminal-deoxynucleoitidyl transferase-mediated nick end labeling (TUNEL) method. H460 and HFL1 cells were seeded in 96-well plates at a density of 2 × 10^4^ cells well^−1^ and grown to ~70% confluence. The cells were incubated with culture medium supplemented with 0, 5, or 10 μM rScyreprocin. Cell apoptosis was detected using In Situ Cell Death Detection Kit at 24 and 48 h post-incubation following the manufacturer’s instruction. The samples were observed and imaged by a Zeiss LSM780 UV-NLO confocal microscope (Carl Zeiss Microscopy GmbH, Oberkochen, Germany).

### 4.9. Cell Immunofluorescent Labeling Assay

H460 and HFL1 cells were seeded in 96-well plates at a density of 2 × 10^4^ cells well^−1^ and incubated overnight. The cells were then incubated with rScyreprocin (0 and 1 μM) for 24 h, fixed in 4% (*w*/*v*) phosphate-buffered paraformaldehyde at 4 °C for 20 min, and permeabilized with 0.1% (*v*/*v*) Triton X-100 for 10 min. The organelles were labeled with corresponding fluorescent probes following the manufacturer’s instructions. In brief, the cells were rinsed with HBSS, incubated in 5% (*w*/*v*) BSA (prepared in HBSS) for 3 h at room temperature, and incubated with scyreprocin antibody (1:1000) overnight at 4 °C. Samples were rinsed with Tris-buffered saline Tween-20 (TBST: 20 mM Tris (pH 7.4), 150 mM NaCl, 0.1% (*v*/*v*) Tween-20) and incubated with Goat anti-Rabbit IgG (H+L) secondary antibody (Dylight 633) (1:1000) for 2 h. The cells were washed with TBST and stained with 1 μg mL^−1^ DAPI for 10 min. The cells were observed by a Zeiss LSM780 UV-NLO confocal microscopy.

### 4.10. Analysis of Mitochondria Membrane Potential (MMP)

H460 and HFL1 cells were seeded in 96-well plates at a density of 2 × 10^4^ cells well^−1^. After 10 h of incubation, rScyreprocin (0, 1, and 4 μM) was added. After 24 h, cells were assessed for the MMP using a lipophilic probe JC-1 according to the manufacturer’s instructions (Solarbio, Beijing, China) and observed with a Zeiss LSM780 UV-NLO confocal microscope.

### 4.11. Measurement of Intracellular Reactive Oxygen Species (ROS) and Intracellular Ca^2+^ Concentration

H460 and HFL1 cells were pre-loaded with DCFH-DA probes or Fluo-4/AM according to the manufacturer’s instruction, respectively. The cells were treated with 0, 1, 2, 4, and 8 μM rScyreprocin. At 2, 8, 10, 24 and 48 h after incubation, the DCFH-DA and Fluo-4/AM fluorescence (*n* = 3) of the samples were analyzed by a microplate reader (TECAN GENios).

### 4.12. Western Blotting Assay

H460 and HFL1 cells were treated with 0 and 1 μM rScyreprocin for 24 h. Cell lysates from four wells were pooled into one for examination. Experiments were carried out on three different occasions (*n* = 3). Samples were subjected to sodium dodecyl sulfate-polyacrylamide gel electrophoresis (SDS-PAGE), transferred to a polyvinylidene difluoride membrane (Millipore Coroperation, Darmstadt, Germany), and analyzed by standard Western blotting protocol. The blot images were obtained by a Tanon™ 5200CE Chemi-Image System (Tanon, Shanghai, China) and analyzed using ImageJ Software (National Institutes of Healthcare, Bethesda, MD, USA).

### 4.13. Extraction of Proteins from Different Cell Compartments

H460 and HFL1 cells were treated with rScyreprocin (0 and 1 μM) for 24 h, and a ProteoExtract^®^ Subcellular Proteome Extraction Kit was applied to extract the cytosolic, membrane, nucleic, and cytoskeletal fractions. Experiments were carried out on three different occasions (*n* = 3). A Qproteome™ Mitochondria isolation Kit was applied to extract mitochondrial and cytosolic fractions from H460 and HFL1 cells. The samples (4 μg) were subjected to Western blotting analysis.

### 4.14. Assessment of In Vivo Anticancer Activity of Rscyreprocin

All experiments involving animals were approved by the Laboratory Animal Management and Ethics Committee of Xiamen University (XMULAC20200030). H460 cells (1 × 10^6^) were injected subcutaneously (s.c.) on the right flanks of male athymic specified pathogen-free (SPF) BALB/c nude mice (18–22 g). Treatment began when the xenograft size reached approximately 100 mm^3^. The mice were treated with 50 μL of PBS, rScyreprocin (1.8 and 3.6 mg kg^−1^), and CDDP (3 mg kg^−1^) by intratumor injection, respectively. Each experimental group contained six mice (*n* = 6). The rScyreprocin injections were performed every 3 days, while the CDDP injections were given every 6 days. The body weight and xenograft size were measured every 3 days.

### 4.15. H&E, TUNEL, and Immunohistochemical Staining

On day 28, the mice were sacrificed. The xenografts were collected, weighed, and sectioned. The specimens were subjected to H&E staining and TUNEL staining (In Situ Cell Death Kit (POD)) following the manufacturer’s instruction. For Ki-67 and CD31 staining, the sections were deparaffinized, rehydrated, and subjected to antigen retrieval. Ki-67 and CD31 antibodies were used as primary antibodies, respectively. The specimens were visualized using UltraSensitive™ SP IHC Kit (MXB Biotechnologies, Fuzhou, China). All specimens were imaged using an optical microscope. The percentage of TUNEL^+^, Ki-67^+^, and CD31^+^ area was analyzed by Image J software (National Institutes of Health, Bethesda, MD, USA).

### 4.16. Statistical Analysis

Data are presented as means ± standard deviations (SD). For the cell viability assay, colony formation assay, and densitometric analysis, two-way analysis of variance (ANOVA) with Bonferroni post-test was applied. For the JC-1 assay and tumor weight analysis, one-way ANOVA with Tukey post-test was applied. Statistical analyses were performed using GraphPad Prism Software (version 5.01; GraphPad Software Inc., San Diego, CA, USA), with a confidence level of 95% being considered to be statistically significant.

## Figures and Tables

**Figure 1 ijms-23-05500-f001:**
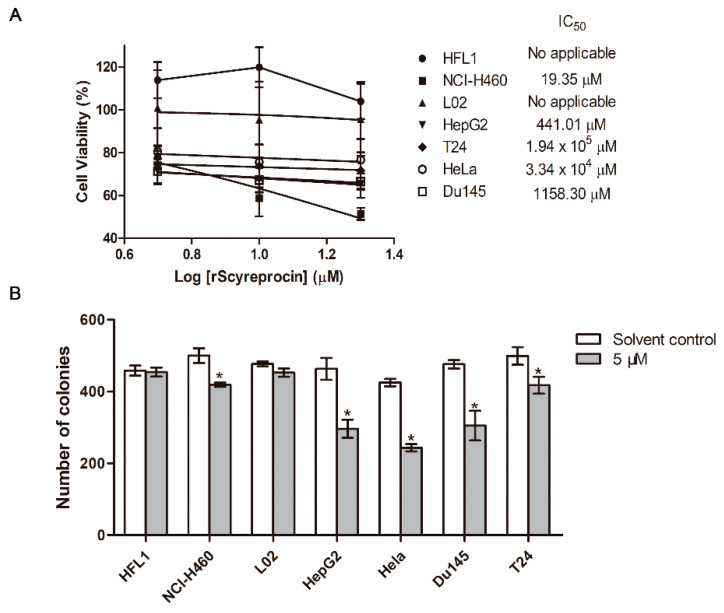
Anticancer activity of recombinant scyreprocin (rScyreprocin). (**A**) Effect of rScyreprocin on cell viability. Cells were incubated with various concentrations of rScyreprocin for 24 h; the cell viability was assessed by MTS method (*n* = 3). (**B**) Effect of rScyreprocin on colony formation ability. Cells were incubated with 5 μM rScyreprocin for 48 h and grown in rScyreprocin-free culture medium for another 12 days; colonies that contained > 50 cells were then counted and analyzed (*n* = 3). Data in (**A**,**B**) are presented as means ± standard deviations (SD). * *p* < 0.05 (two-way ANOVA, Bonferroni post-tests).

**Figure 2 ijms-23-05500-f002:**
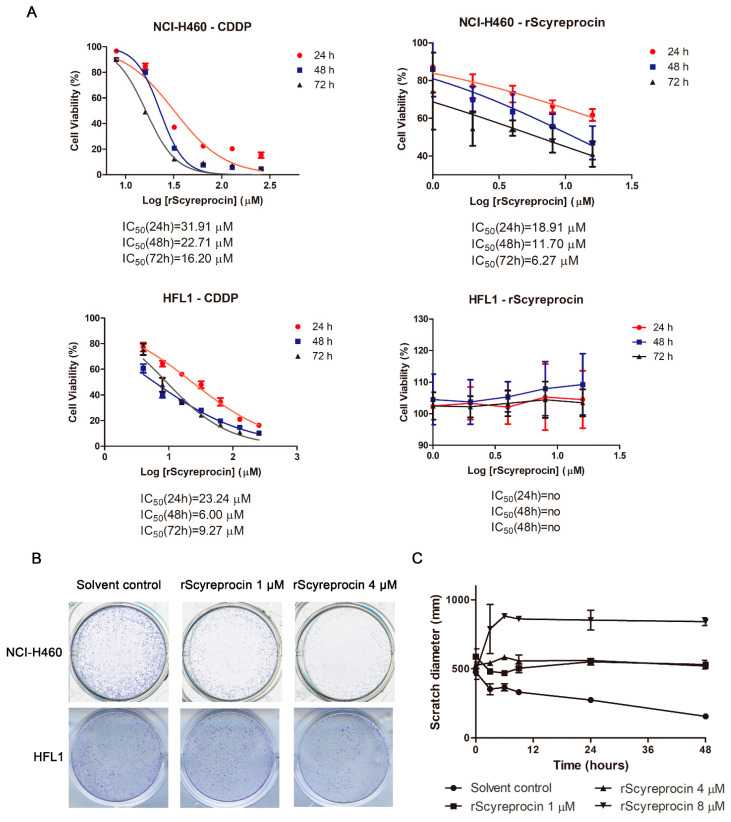
Recombinant scyreprocin (rScyreprocin) inhibited proliferation of H460 cells but not HFL1 cells. (**A**) Effects of rScyreprocin on H460 and HFL1 cell viability. Cells were incubated with various concentrations of rScyreprocin or CDDP; cell viability was determined by MTS method at different time points (*n* = 3). (**B**) Effect of rScyreprocin of H460 and HFL1 colony formation ability. Cells were treated with rScyreprocin for 48 h and cultured in rScyreprocin-free medium for another 12 days. Samples were fixed and stained with crystal violet before microscopic examination. (**C**) Inhibitory effect of rScyreprocin on H460 cell migration. Scratches of H460 were treated with various concentrations of rScyreprocin; the shortest distances between the edges were measured and analyzed at different time points (*n* = 3). Data in (**A**,**C**) are presented in means ± standard deviations (SD).

**Figure 3 ijms-23-05500-f003:**
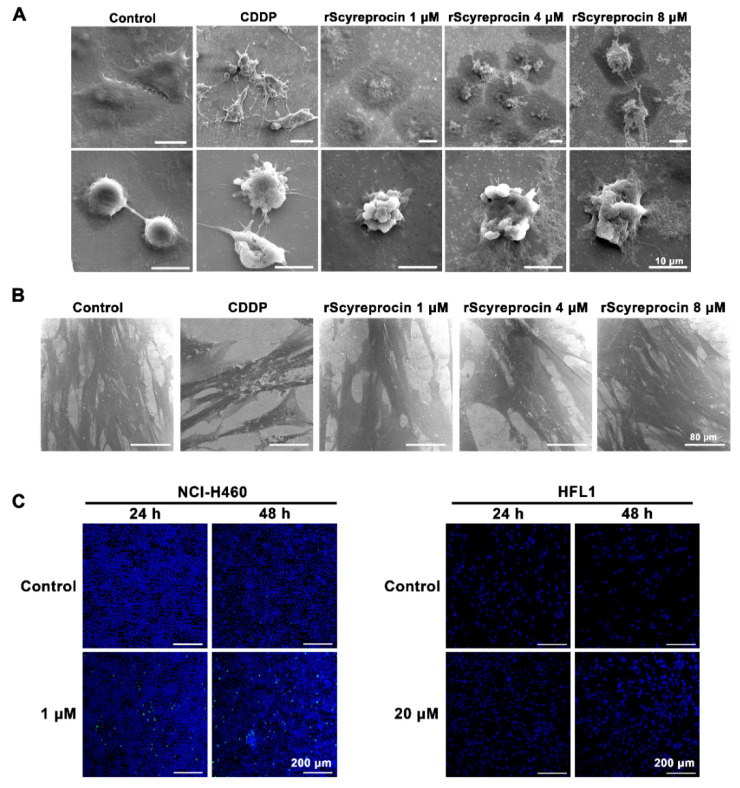
Recombinant scyreprocin (rScyreprocin) induced apoptosis and cell membrane destruction of H460 cells but not HFL1 cells. (**A**) Effect of rScyreprocin on H460 cell morphology. Cells were treated with rScyreprocin (0, 1, 4, and 8 μM) or cisplatin (CDDP) for 24 h and observed for morphological changes by a scanning electron microscope (SEM) (scale bar = 10 μm). (**B**) Effect of rScyreprocin on HFL1 cell morphology. HFL1 cells were incubated with rScyreprocin (0, 1, 4, and 8 μM) or CDDP (32 μM) for 24 h and observed by a scanning electron microscopy (scale bar = 80 μm). (**C**) Apoptotic effect of rScyreprocin on H460 and HFL1 cells. Cells were incubated with rScyreprocin and evaluated for apoptosis using TUNEL staining. Nuclei were visualized by DAPI staining (blue); a positive signal (green) indicating apoptosis was detected in rScyreprocin-treated H460 cells but not in HFL1 cells (scale bar = 200 μm).

**Figure 4 ijms-23-05500-f004:**
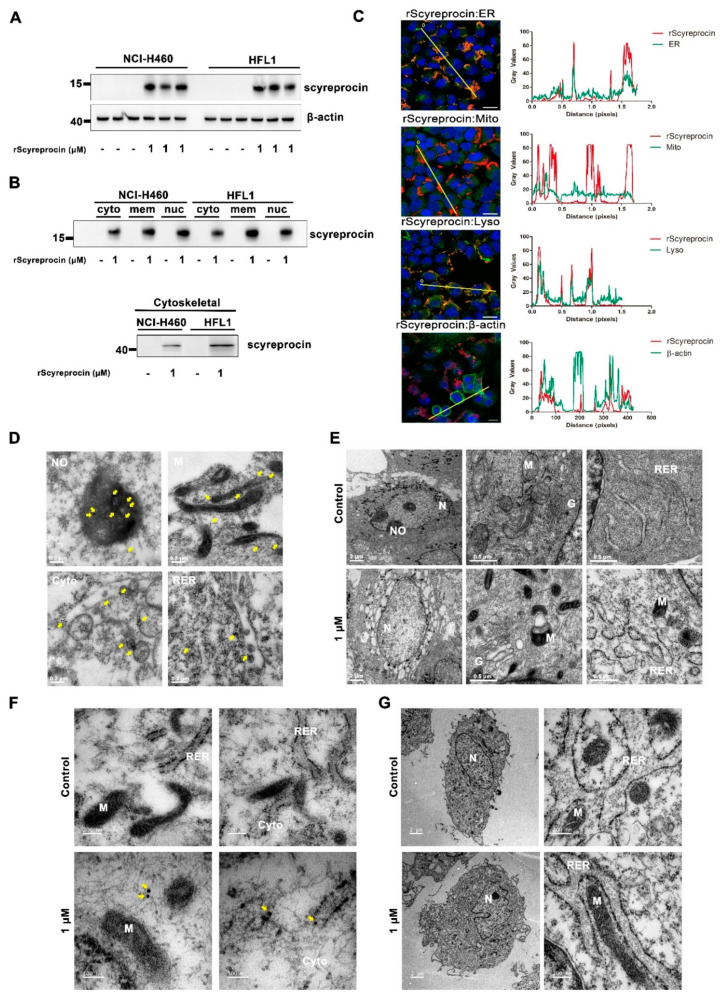
Recombinant scyreprocin (rScyreprocin) penetrated cell membrane and targeted and damaged organelles of H460 cells. (**A**) Detection of rScyreprocin penetrated into H460 and HFL1 cells. Cells were incubated with rScyreprocin (0 and 1 μM) for 24 h, and analyzed by Western blot assay using scyreprocin antibody (*n* = 3). See also Appendix A. (**B**) Detection of rScyreprocin in different cell fractions. H460 and HFL1 cells were treated with rScyreprocin (0 and 1 μM) for 24 h. Proteins from cytosolic (cyto), membrane (mem), nucleic (nuc), and cytoskeletal fractions were extracted and analyzed (4 μg) by Western blot assay using scyreprocin antibody. (**C**) Intracellular distribution of rScyreprocin in H460 cells. Cells were treated with rScyreprocin (0 and 1 μM) for 24 h and subjected to organelle staining and cell immunofluorescence assay. The red color indicates the localization of rScyreprocin and the green color indicates the organelles, including endoplasmic reticulum (ER), mitochondria (Mito), lysosomes (Lyso), and cytoskeleton (Actin). The yellow color represents the co-localization of rScyreprocin and organelles. The blue color indicates the nucleus (scale bar = 20 μm). (**D**) Subcellular localization of rScyreprocin in H460 cells. After a 24 h treatment with rScyreprocin (1 μM), rScyreprocin in NCI-H40 cells was labeled by immunocolloidal gold technique and observed by a transmission electron microscope (TEM). Positive signals (yellow arrows) were found in nucleolus (NO), mitochondria (M), cytoplasm (cyto), and rough endoplasmic reticulum (RER). (**E**) Effects of rScyreprocin on organelle structure. H460 cells were treated with rScyreprocin (0 and 1 μM) for 24 h. Overt morphological changes of nucleolus (NO), mitochondria (M), rough endoplasmic reticulum (RER), and Golgi apparatus (G) were observed by TEM observation. (**F**) Subcellular localization of rScyreprocin in HFL1 cells. After a 24 h treatment with rScyreprocin (1 μM), rScyreprocin in HFL1 cells was labeled by immunocolloidal gold technique and observed by a TEM. Positive signals (yellow arrows) were found in cytoplasm (cyto), but only a few positive signals were located near organelles. (**G**) Effects of rScyreprocin on organelle structure. HFL1 cells were treated with rScyreprocin (0 and 1 μM) for 24 h before TEM observation. No morphological change was found in rScyreprocin-treated cells compared to control group.

**Figure 5 ijms-23-05500-f005:**
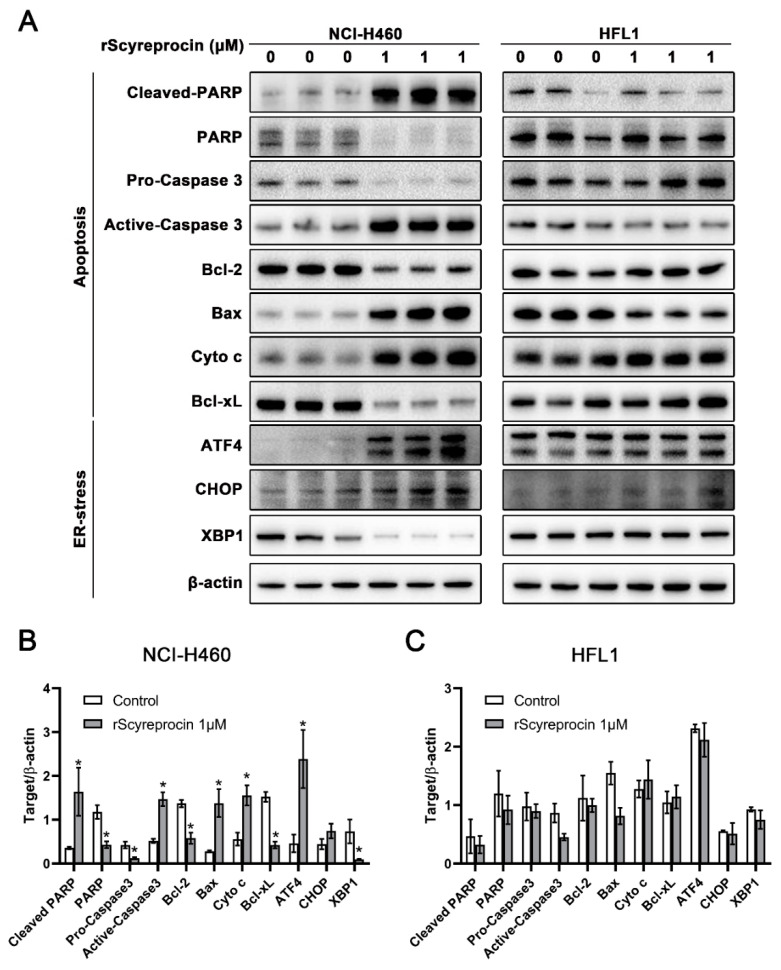
Recombinant scyreprocin (rScyreprocin) induced changes in expression levels of key proteins in pathways associated with apoptosis and ER stress. H460 and HFL1 cells were treated with rScyreprocin (0 and 1 μM) for 24 h. Levels of apoptosis-related and ER stress-related proteins in whole cell lysate (*n* = 3) were determined by Western blotting (**A**). Data were quantified using Image J Software and are presented in (**B**,**C**). Data are presented in means ± standard deviation (SD). In (**B**,**C**), data were normalized to the value of β-actin and analyzed by Student’s *t*-tests. (* *p* < 0.05).

**Figure 6 ijms-23-05500-f006:**
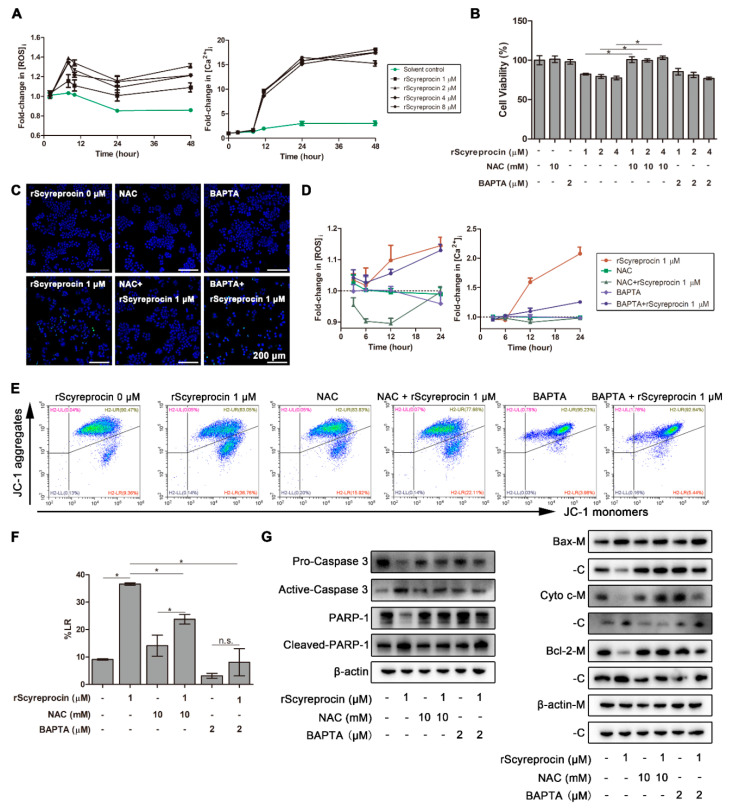
Recombinant scyreprocin (rScyreprocin) induced intracellular reactive oxygen species (ROS_i_) production, caused mitochondria dysfunction, and led to apoptosis of H460 cells. (**A**) Fold changes in intracellular Ca^2+^ and ROS levels of rScyreprocin-treated H460 cells (*n* = 3). Cells were treated with rScyreprocin. Data were collected at different time points, and normalized to the reads obtained at 3 h. (**B**) Inhibitory effect of rScyreprocin on NAC pretreated H460 cells. Cells were incubated with rScyreprocin for 24 h in the presence or absence of prior 1 h incubation with 10 mM NAC, and were evaluated for cell viability (*n* = 3). (**C**) Apoptotic effect of rScyreprocin on NAC and BAPTA pretreated H460 cells. Cells were treated with 1μM rScyreprocin for 24 h in the presence or absence of prior 1 h incubation with NAC or BAPTA, respectively. In situ cell death was determined by TUNEL assay. (**D**) Effect of rScyreprocin on intracellular Ca^2+^ and ROS levels. Samples were prepared as described in (**C**) and evaluated for fold changes in intracellular ROS and Ca^2+^ levels, respectively (*n* = 3). (**E**) Effect of rScyreprocin on mitochondrial membrane potential of H460 cells. Samples were prepared as described in (**C**), subjected to JC-1 staining, and analyzed with flow cytometry (*n* = 3). (**F**) Statistical analysis of the data presented in (**E**). (**G**) Expressions of apoptosis-related proteins in rScyreprocin-treated H460 cells. Samples were prepared as described in (**C**). Expressions of apoptosis-related proteins in whole cell lysate (left panel) and expressions of Bax, cytochrome c, and Bcl-2 in mitochondrial (M) and cytosolic (C) fractions (right panel) were determined. The experiments were carried out in three different occasions (*n* = 3) and quantified using Image J Software. Blots and quantification data of the replicates are presented in Appendix A. Data are presented as means ± standard deviation (SD). In (**A**), data were normalized to the value of solvent control at 3 h. In (**B**), data were normalized to the control group and analyzed by one-way ANOVA with Tukey post-tests. In (**D**), data are presented as means ± SD, and data were normalized to values of control groups at each time point. In (**F**), data were analyzed by one-way ANOVA with Tukey post-tests (* *p* < 0.05; n.s.—non-significant).

**Figure 7 ijms-23-05500-f007:**
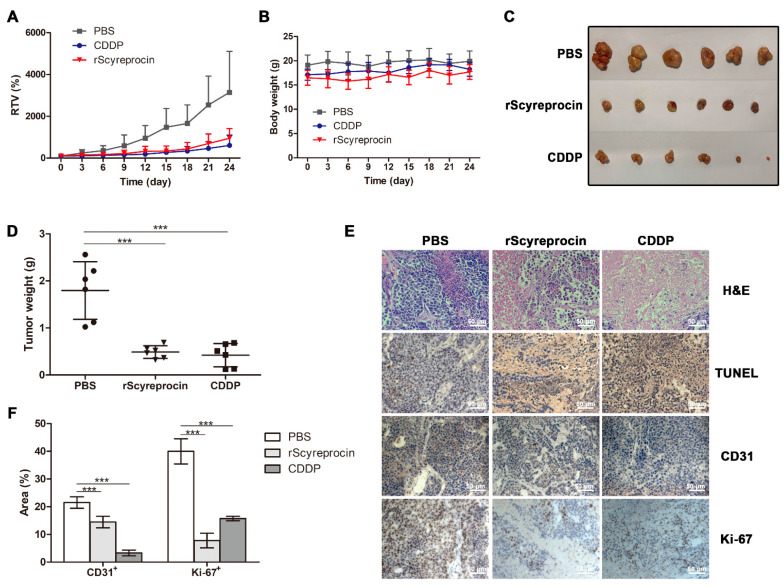
Recombinant scyreprocin (rScyreprocin) suppressed tumor growth in vivo. (**A**) Relative tumor volume (RTV) in different treatment groups. Cisplatin (CDDP, 3 mg kg^−1^), rScyreprocin (1.8 mg kg^−1^), and PBS (control) were given by intratumor injections; tumor volumes were measured every 3 days (*n* = 6). (**B**) Body weight of mice in different treatment groups. The body weights of the mice were measured every 3 days (*n* = 6). (**C**) Tumor tissues dissected from mice with different treatments. (**D**) Tumor weight of each group (*n* = 6). (**E**) H&E, TUNEL, Ki-67, and CD31 staining of tumor tissue sections obtained from mice with different treatments (scale bar = 50 μm). (**F**) Quantification of percent of Ki-67^+^ and CD31^+^ areas of each group (*n* = 5). Data are presented as means ± standard deviation (SD). In (**A**), data were normalized to the initial tumor volume measured at day 0. In (**B**), data were normalized to the control group and analyzed by one-way ANOVA with Tukey post-tests (*** *p* < 0.0001). In (**F**), data were analyzed by two-way ANOVA with Bonferroni post-test (*** *p* < 0.0001).

## Data Availability

Not applicable.

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
