# Peer review of "The Anticancer Activity Conferred by the Mud Crab Antimicrobial Peptide Scyreprocin through Apoptosis and Membrane Disruption"

_ijms, 2022, doi:10.3390/ijms23105500_

Round 1

Reviewer 1 Report

The manuscript by Yang et al. is dedicated to investigation of anti-cancer properties of antimicrobial peptide Scyreprocin.

The article is of high enough quality, well-written, the experimental design is appropriate, main conclusions are supported by experimental results, there are only some minor issues to answer before article acceptance:

  1. Fig 1A, Fig 2 A, Fig S2 - consider making a curve fit instead of columns and compare the IC50 of Scyreprocin on different cancer/normal cell lines.
  2. Fig 2B - It seems that HFL1 forms a monolayer instead of the colonies - am I right? Could you explain that?
  3. Fig 3C - the image of H460 cells is of poor quality, please replace.
  4. Fig. 4C - you should consider performing colocalization analysis (Costes in Zen or ImageJ)
  5. It would be interesting to know whether Scyreprocin inhibits the growth of CDDP-resistant cells such as A549? Even in vitro data will be extremely important
  6. Please provide all Abs Cat #s in M&M or in supplementary files. The original images of WBs are of high quality.

In general, the article can be considered after minor revisions.

Reviewer 2 Report

AA aim to demonstrate anticancer activity of the antimicrobial peptide Scyreprocin. They show that recombinant Scyreprocin (rS), in vitro, induces membrane damage and apoptosis of cancer cells without affecting two immortalized normal cells; in vivo, that peptide strongly reduces tumor growth, with an efficiency comparable to cisplatin. The study is interesting, results on peptide effects are also supported by previous literature. However, in my opinion, AA need to address the following questions and concerns.

General comments

  • In some figure legends: why AA write “Cells were co-incubated with rScyreprocin” ? Cells were incubated with something else beside rS? If not, please correct in “incubated”.
  • In some Figure legends, AA indicate n=3, that would mean that they performed each experiment only once in triplicates, but the experiments should be repeated at least three times and means should be the mean of three experiments and not the mean of triplicates of a single experiment. Please, elucidate this point and possibly add/correct.

Specific comments

Introduction, page 2, Lane 14: Scyreprocin is a cationic AMP that was first identified…

- A very recent publication of same AA on the same peptide on IJMS (2022) could be added.

Paragraph 2.1:

  • AA should specify, in the text, from which cancers (or normal tissues) ALL cell lines are derived.
  • In the text, authors should specify that rS effect on HeLa cells is not statistically significant.
  • 1: it would be better to compare normal and cancer cells derived from same tissues in a graph and the other cancer cell lines in a separate graph. Cancers from different tissues have different characteristics.
  • 1, legend: Please correct: Data in (A) and (B) are presented as means ± standard deviations
  • Figure 2A shows cell viability, not cell proliferation, please correct in the text.
  • Figure 2A: The experiments to determine IC50 for rS should be shown and compared with same experiments with CDDP in the same figure (CDDP is in supplementary fig).
  • Figure 2B: why there is so much difference in the staining of H460 cells and HFL1 cells incubated with solvent? Please, provide comparable pictures (not so much darker for HFL1).

Paragraph 2.2:

- lane 8: “These results suggested that rScyreprocin exerted its anticancer activity by both inducing

cell apoptosis and disrupting cell membrane integrity.”. Disruption of cell membrane integrity means that there is necrosis, please correct/add.

- Figure 3B: it is indicated only concentration and not “rScyreprocin”

- Fig.3 supplementary: it is unexpected that scratches appear larger instead of closing in cells treated with rS. there is an explanation? Please, comment this point. However, in order to compare migration of different cells, quantitative migration assay in Boyden chambers is more appropriate.

Paragraph 2.3:

- Fig.4A: It should be indicated the concentration of rS, as indicated in the legend, instead of 000 and 111, or are they triplicates of 1mM (in this case, AA have to correct in the legend). How many times the experiment has been repeated?

- Figure 4B: Is it possible to add markers specific for the different subcellular fractions?

Paragraph 2.4:

- Second  sentence, please correct: “while a remarkable intracellular Ca2+ concentration elevation between 8 and 12 h after rScyreprocin treatment was observed (Figure 6A)”.

- NAC and BAPTA: AA should indicate full names and, importantly, their functions /effects, in order to elucidate the meaning of their use.

- Figure 5A: Triplicates are very variable for HFL1, they should be homogeneous, as for H460, in this condition is very hard to state that levels of analyzed proteins are not affected by rS (despite the mean); also, there is a clear and unexpected decrease in BAX, that should be at least commented.

- What is MMP? please write full names

- Figure 6: please add  rScyreprocin beside the concentration.

- Figure 6E-F: BAPTA effect appear strong in E but it is not evident in F, please comment

Discussion: “In this study, rScyreprocin activated ER stress markers, ATF4 and CHOP, indicating that rScyreprocin induced ER stress in H460cells (Figure 5). The exact role of Ca2+ signaling in rScyreprocin-induced ER stress remains to be studied”. Please, correct.

Finally, a crucial point should be better discussed: why is the peptide specific for cancer cells, as AA state.

Round 2

Reviewer 2 Report

 AA fully addressed questions and concerns.